# Effects of the COVID-19 Pandemic on the Interest of Google Queries in Cancer Screening and Cancers: A Retrospective Study

**DOI:** 10.3390/cancers15030617

**Published:** 2023-01-19

**Authors:** Mikołaj Kamiński, Piotr Skrzypczak, Rafał Staszewski, Magdalena Roszak

**Affiliations:** 1District Hospital in Kościan, Department of Rheumatology, 64-000 Kościan, Poland; 2Department of the Treatment of Obesity and Metabolic Disorders, and of Clinical Dietetics, Poznań University of Medical Sciences, 60-569 Poznań, Poland; 3Department of Thoracic Surgery, Poznan University of Medical Sciences, Szamarzewskiego St. 62, 60-569 Poznań, Poland; 4Department of Hypertension, Angiology and Internal Diseases, Poznan University of Medical Sciences, 61-848 Poznań, Poland; 5Department of Computer Science and Statistics, Poznan University of Medical Sciences, Rokietnicka St. 7, 60-806 Poznań, Poland

**Keywords:** google trends, cancer, screening, COVID-19, SARS-CoV-2, pandemic, global interest

## Abstract

**Simple Summary:**

The COVID-19 pandemic caused the introduction of restrictions to reduce human migration and gatherings. One of the highly harmful side effects was the disturbances in medical services and continuity of health care. A broad group of oncological patients wishing to undergo screening may search for medical information on the Internet. Using Google Trends statistics, we analyzed the effects of the COVID-19 pandemic on the interest in cancers and their screenings. Our study aimed to comprehensively compare global interest in cancers and their screenings in 2020–2021. The interest of Google users in cancer screenings increased in 2020–2021 compared to 2015–2019, but the growth was less dynamic than expected. The interest in many cancers during the COVID-19 pandemic was significantly lower than in the prepandemic period, especially during March and April 2020. A loss of interest in cancer screenings may delay the diagnosis of malignancies and worsen the long-term outcomes.

**Abstract:**

The COVID-19 pandemic disrupted cancer screening programs and care for individuals with malignancies. We aimed to analyze the effects of the COVID-19 pandemic on the interest of Google users in cancers and their screenings. We collected data from Google Trends (GT) from 1 January 2015 to 31 December 2021 worldwide for nine topics representing cancer screening and the HPV vaccine and for 33 topics representing malignancies. We performed a secular analysis comparing the prepandemic (2015–2019) and pandemic (2020–2021) period. We performed forecasting analysis on the prepandemic timeline to assess interest in the analyzed topics if the pandemic hadnot occurred.The actual interest in most of the analyzed topics was significantly lower than in the forecasted trend. Interest in 6 of the 9 topics representing cancer screening and 3 of the 33 topics representing cancer was higher during the pandemic than in the prepandemic period. The interest of Google users in cancer screenings increased in 2020–2021 compared to 2015–2019, but the growth was less dynamic than expected. The interest in many cancers during the pandemic was significantly lower than in the prepandemic period, especially during March and April 2020. The lower interest in cancers and their screenings may delay the diagnosis and worsen the long-term outcomes.

## 1. Introduction

The unexpected COVID-19 pandemic made many governments introduce numerous restrictions to reduce human migration and public gatherings, limiting the virus’s global spreading [1]. The most severe restrictions took the form of full and moderate lockdowns [2]. Unfortunately, such dynamic policies had adverse psychosocial and economic effects, as analyzed by numerous authors [1,2,3].

One of the highly harmful consequences of lockdowns was the disturbances in medical services and the continuity of health care [4]. The COVID-19 pandemic has disturbed healthcare systems and the continuity of care globally. Significant delays and limitations were noticeable in primary care admissions and more advanced treatments such as oncologic surgery or chemotherapy [4,5,6,7,8]. Moreover, the lockdown disrupted cancer screening programs in many countries, which could have delayed the diagnosis of malignancies [9,10,11].

For 70% of cancer patients, the Internet is the first source of information [12]. A broad group of patients suffering from malignant tumors or wishing to undergo screening may search for medical information on the Internet, which most generations use daily [13]. Therefore, Internet traffic may reflect the interest of the population in health issues [14].

Search engine query analysis estimates the interest in various medical issues and may associate some epidemiological data worldwide [15]. We hypothesize that the analysis of searches on cancers and their screenings may reveal an under-researched phenomenon related to the dynamics of interest in this topic. Google is one of the most popular search engines globally, acquiring 90 percent of the global market in 2019 [16]. For this reason, Google search statistics presented in Google Trends (GT; https://trends.google.com/trends/, accessed on 26 February 2022) are commonly utilized for search engine query analyses [14].

GT has found applications in assessing such issues as global interest in different cancer treatment methods or primary symptoms [17,18]. However, no paper has comprehensively compared global interest in cancers and their screenings in 2020–2021.

Here, we aimed to analyze the effects of the COVID-19 pandemic on the interest of Google users in cancers and their screenings.

## 2. Materials and Methods

We retrospectively analyzed freely available data provided by GT (https://trends.google.com/trends/, accessed on 26 February 2022). For this reason, the project did not require Ethical Board approval.

### 2.1. Data Collection 

GT presents a sample of Google search statistics [19]. The intensity is expressed as therelative search volume (RSV). The RSV ranges from 0 to 100, with 0 representing a complete lack of interest (0%) and 100 corresponds to the peak of popularity (100%) [14]. The RSV is for a given period and location. GT has collected statistics since January 2004.

GT may match a given input and “search term” or “topic”. Search terms are the exact phrases typed into the GT search engine. GT may propose a matched topic for a given search term. Topic statistics collect all topic-related queries independently of the language of Google users. Topics allow for the comparison of the given terms between all countries, while search terms cannot [20,21]. For example, the search term “apple” will generate Google statistics for the literal term; thus, the RSV will be the highest in English-speaking countries. If the topic “Apple” is chosen, all queries related to the topic, regardless of language, will be included in GT statistics.

We collected data from GT from 1 January 2015 to 31 December 2021. The data point was the following months. 

Firstly, we searched for cancer-screening-related topics. M.K. and P.S.jointly created an initial list of search terms: “cancer screening”, “breast cancer screening”, “mammography”, “colorectal cancer screening”, “colonoscopy”, “fecal occult blood test”, “cervical cancer screening”, “cervical cytology”, “Pap smear”, “prostate cancer screening”, “prostate-specific antigen”, and “lung cancer screening”. Moreover, we added the search term “HPV vaccine”, which is recommended as the primary prophylaxis of cervix uteri, anal, and oropharynx cancers [22,23]. We matched the following topics: “Breast cancer screening”, “Cancer screening”, “Colon Cancer”, “Screening”, “Colonoscopy”, “Fecal occult blood”, “HPV vaccine”, “Mammography”, “Pap test”, “Prostate-specific antigen”, and “Prostate cancer screening”. 

Further, we used the names of cancers considered in Global Cancer Statistics GLOBOCAN 2018: lung cancer, breast cancer, prostate cancer, colon cancer, stomach cancer, liver cancer, rectum cancer, esophagus cancer, cervix uteri cancer, thyroid cancer, bladder cancer, non-Hodgkin’s lymphoma, pancreas cancer, leukemia, kidney cancer, corpus uteri cancer, lip/oral cavity cancers, brain/nervous system malignancies, ovary cancer, melanoma of the skin, gallbladder cancer, larynx cancer, multiple myeloma, nasopharynx cancer, oropharynx cancer, hypopharynx cancer, Hodgkin lymphoma, testis cancer, salivary glands cancer, anus cancer, vulva cancer, Kaposi sarcoma, penis cancer, mesothelioma, and vagina cancer [24]. We did not include nonmelanoma skin cancers. We matched the following topics: “Anal cancer”, “Bladder cancer”, “Brain Cancer” (matched “[central] nervous system maglinancy”), “Breast cancer”, “Cervical cancer”, “Colorectal cancer”, “Esophageal cancer”, “Gallbladder cancer”, “Head and neck cancer” (matched: “nasopharynx cancer”, “oral cancer”), “Hodgkin’s lymphoma”, “Kaposi’s sarcoma”, “Kidney cancer”, “Laryngeal cancer” (matched:),“Leukemia”, “Liver cancer”, “Lung cancer”, “Melanoma”, “Mesothelioma”, “Multiple myeloma”, “Nasopharyngeal carcinoma” (matched: “nasopharynx cancer”), “Non-Hodgkin lymphoma”, “Oral cancer” (matched: “oral cancer”), “Ovarian cancer”, “Pancreatic cancer”, “Penile cancer”, “Prostate cancer”, “Salivary gland tumour” (matched: “salivary glands cancer”), “Stomach cancer”, “Testicular cancer”, “Thyroid cancer”, “Uterine cancer” (matched: “corpus uteri cancer”), “Vaginal cancer”, and “Vulvar cancer”.

We set the region to “Worldwide” and included all categories of research. All chosen topics were typed separately in GT, and we collected data about interest over time. We reported all details on the search strategy according to the Nuti protocol (Appendix A) [14].

### 2.2. Data Analysis

We performed data manipulations and statistical analyses using the R-programming language (version 3.6.1.; Vienna, R Project). We divided the timeline into two periods: prepandemic (1 January 2015–31 December 2019) and during the pandemic (also further called “actual”) (1 January 2020–31 December 2021). We used a prepandemic timeline to perform forecasting for the following 24 months (1 January 2020–31 December 2021). We fitted the time series for the years 2015–2019 to an exponentialsmoothing state-space model with BoxCox transformation, autoregressive-moving average errors, and trend and seasonal components (TBATS) using the forecast package of R [25]. If the time trend does not present significant seasonality, the TBATS model will include only secular trends in the forecasted timeline. Previous studies suggested that topics with low relative interest more often present irregular variations of the RSV and a lack of significant seasonal trends [20,26,27].

We expressed numerical RSV values as the median (interquartile range). We calculated the RSV difference between the sum of the actual RSV during the COVID-19 pandemic and the sum of the forecasted time trends through 24 data points (24 months). We performed a Mann–Whitney U test to compare a) the prepandemic RSV vs. actual RSV during thepandemic, and b) the actual RSV vs. forecasted RSV during the pandemic. The *p*-value threshold was set to 0.05. We used the ggplot2 package to visualize time trends [28].

We performed a sensitivity analysis. We excluded two months of the very beginning of the pandemic (March and April 2020) from the comparison of the actual vs. forecasted RSV during the pandemic. These two months were chosen based on the literature that suggested that the interest of Google users in medical-related topics decreased during the early months of the pandemic [29,30,31,32]. Furthermore, we excluded months with unusually high interest that could be related to, e.g., the death of a famous person due to a specific malignancy [33,34].

The dataset is presented in Appendix A

## 3. Results

We collected data for *n* = 9 representing cancer-related screenings and “HPV vaccine” and for *n* = 33 topics representing malignancies. Overall, the raw difference between the actual and forecasted RSV for all topics but “HPV vaccine” was negative, which suggests a decrease in interest than the forecasted trend based on GT data from 2015–2019 (Table 1 and Table 2, Figure 1, Figure 2 and Figure 3). The sorted raw difference is presented in the Appendix A.

We found interest in topics representing 6 of the 9 cancer-related screenings: “Cancer screening”, “Colon Cancer screening”, “Colonoscopy”, “Fecal occult blood”, Mammography”, “Prostate-specific antigen”,and in“HPV vaccine”. Additionally, interest in“Mesothelioma”, “Prostate cancer”, and “Vulvar cancer”was higher during the COVID-19 pandemic than in the years 2015–2019 (Table 1 and Table 2). However, interest was lower for the topics representing 16 of the 33 malignanciesthan in prepandemic period: “Brain cancer”, “Breast cancer”, “Cervical cancer”, “Gallbladder cancer”, “Kaposi’s sarcoma”, “Kidney cancer”, “Leukemia”, Liver cancer”, “Lung cancer”, “Melanoma”, “Oral cancer”, “Ovarian cancer”, “Penile cancer”, “Testicular cancer”, “Thyroid cancer”, and “Uterine cancer”.

We found that the actual RSV was significantly lower than the forecasted RSV for “Cancer screening”, “Colonoscopy”, “Mammography”, “Pap test”, and “Prostate cancer screening” (Figure 1, Table 1). Moreover, actual interest in all topics representing malignancies except “Colorectal cancer”, “Kaposi’s sarcoma”, and “Nasopharyngeal carcinoma” was also significantly lower than forecasted(Figure 2 and Figure 3, Table 2). Our analysis did not find any example of significantly higher actual interest than forecasted interest.

The data presented several peaks of interest during the COVID-19 pandemic for the actual RSV of the topics “Colon Cancer Screening” (August 2020), “Colorectal cancer” (August 2020), “Hodgkin’s lymphoma” (March 2020), “Mesothelioma” (November 2021), and “Multiple myeloma” (October 2021) (Figure 2 and Figure 3). Many topics revealed a decrease in interest during the first months of the first wave of COVID-19 (March and April 2020).

A sensitivity analysis confirmed differences in the main analysis except for several topics (Appendix A). We found that the exclusion of March and April 2020, as well as unusual peaks of interest, caused interest in “HPV vaccine” to become significantly higher in actual trends than in forecasted trends. Moreover, the RSV in “Multiple myeloma” and “Breast cancer screening” during the COVID-19 pandemic became significantly higher than in the prepandemic period. Finally, there were no significant differences between interest in “Kidney cancer” in the years 2015–2019 and during the pandemic as well as between the actual vs. forecasted RSV for “Salivary gland tumour”.

## 4. Discussion

To our best knowledge, our study is the first to broadly assess the global interest of Google users in cancer and oncological screenings. The main findings of our study are that (1) there was a significant increase ininterest in oncology screening during the COVID-19 pandemic, but the increase was less dynamic than forecasted, and (2) for some cancers, the interest decreased in comparison to 2015–2019, but even if it increased, it was less than expected.

When analyzing the GT data, we couldcalculate the RSV for the actual and forecasted periods. When evaluating the data, even if we observed the growth of the actual RSV, we should have also adjusted this value to the forecasted RSV. Whenassessing the trends of interest of Internet users, not only is the absolute value of the RSV essential, but the dynamics of its changes are too. In our study, the decreased actual interest compared to the forecasted interest means that the pandemic caused an essential disturbance in the dynamics of the growth of interest in oncological prevention.

It is tempting to hypothesize thatduring the pandemic, patients would use the Internet to seek ways to protect themselves against cancer whentheir access to professional medical services remained limited. However, our results suggest somethingdifferent. We observed an increased interest in issues such as “cancer screenings”, “colonoscopy”, “mammography”, “Pap test”, and “prostate cancer screening”; however, in many cases, it was significantly lower than the forecasted ones. Only interest in the “HPV vaccine” became higher in the actual trends than the forecasted trends; however, thisdid not reach statistical significance. Thisallows for the conclusion that the pandemic caused a disturbance in the dynamics of interest in oncological screenings.

Especially in March and April 2022, the first months of the pandemic, we documented a significant decline in interest in cancer screenings. It seems plausible that this effect could be associated with the national lockdowns, replacing general practitioner visits with teleconsultations, or transferring other medical services into COVID units. The decline in interest in screenings that we observed is consistent with the lower number of screenings performed, according to one of the latest available studies analyzing the number of tests performed for the three most prevalent cancers [35].

During the first peak of the COVID-19 pandemic, delays were noticeable, among others, in obtaining oncological diagnoses and performing elective procedures, not to mention the significant decrease in the number of screening tests performed [36,37,38]. Some authors claim that the frequently exaggerated reports of the increased risk of SARS-CoV-2 infection after hospital admission could make patients less eager for visits and admissions [36]. During the first wave of the pandemic and the beginning of the official lockdown in April 2020, some countries recorded a drop of up to 90% in endoscopies [36]. The authors noted the most significant drops in interest in Spring 2020 for mammographies, PSA tests, colonoscopies, and Pap smears [39].

Snyder et al. analyzed the global search volume for various screening examinations during the first peak of the COVID-19 pandemic. They found that interest in issues such as “colonoscopy,”“mammogram,”“lung cancer screening,” and “pap smear” had decreased during the first wave by more than 70% compared to the prepandemic period [29].

These observations are consistent with epidemiological data. The monthly number of new suspected cancer cases in Poland diminished by over 30 percent during the first lockdown in March 2020. In the USA, adequately conducted screening accounts for 90% of diagnosed prostate cancers [40]. In one of the USA states, the 60% decrease in prostate-specific antigen screening during the pandemic resulted in a significant decrease in diagnosis [41].

The pandemic has shifted cancer-related Internet traffic toward COVID-19 [42,43]. The global increase in Internet search volumes for health topics related to COVID-19 [44] was followed by a significant decrease in interest in cancer screening and symptoms [36,37,38]. The paper assessing the queries in the field of laryngology described an increase in terms typical of COVID-19, followed by a decrease in searches related to cancer issuessuch as “thyroid cancer” [31]. Khosla et al. assessed the interest in urologic conditions. They reported an increased interest in such conditions as erectile dysfunction and a significant decrease in urologic cancers and their symptoms: bladder cancer, hematuria, and prostate or kidney cancer [18].

We found that interest in the most common neoplasms worldwide, i.e., lung cancer, prostate cancer, breast cancer, melanoma, and cervical and ovarian cancer, decreased significantly during the pandemic period. The challenging treatment of these cancers, combined with the lower patient awareness, could potentially lead to poor long-term results and higher mortality [5]. Moreover, the interest in many cancers has decreased, and we did not find any example of actual interest that was significantly higher than the forecasted interest. Furthermore, many topics revealed a decrease in interest during the first wave of COVID-19 (March and April 2020). A loss of interest in cancer screenings and risk factors may increase the symptomatic phase’s detectability and ultimately worsen long-term survival and therapeutic success [45].

This trend was confirmed by a study conducted in the United States, which observed a decline in search interest for all cancer types in the early stage of the pandemic. It should be emphasized that the interest in such common cancers as breast cancer, colorectal cancer, or melanoma has decreased significantly [32].

A study assessing cancer public interest in Canada reported a significant decrease in search volumes for common cancers. A considerable decline was recorded in the initial stage of the pandemic in the Spring of 2020. These study results are consistent with ours: the diminished interest in 2020 affected breast cancer, colon cancer, lung cancer, and prostate cancer [46]. A similar decline in public interest was described in primary genitourinary cancers such as prostate, kidney, and bladder cancer in the Summer of 2020 [47].

We also observed unusual peaks of interest for some topics. Such rapid changes ininterest in a given disease are often associated, for example, with the death of a famous person caused bythat disease. Examples of this are the growing interest in colorectal cancer following the death of Chadwick Boseman in August 2020 [34], the death of Collin Powell from multiple myeloma in October 2021 [48], and the death of Valerie Harper in August 2019 from a brain tumor [49]. Similar results have been reported by other authors who observed comparable RSV values for some oncological queries one year after the COVID-19 outbreak with the year before the pandemic [39]. The cyclical nature of health-related searches, not the pandemic outbreak itself, may explain the periodic increase in interest in cancer-related queries [50]. However, we included two years to limit the effect of seasonal cycling.

Some researchers described that people searching for health information online have a higher chance of successful screening [50]. Moreover, some papers report the association between Google search activity and the mortality rate [51] in some malignancies. The availability of decent-quality online content on cancer awareness and cancer screenings may be crucial. Our research includes analyses of all cancers and oncological screenings for the entire globe. This allows for a generalization and a broad analysis of possible clinical implications. The further practical implications should also be appropriate information campaigns on the occurrence of the most common neoplasms and their prevention. Oncology should be of particular concernto prioritize elective surgery pathways in the most time-related cancers, and improving the capacity of surgical centers would be an essential step in mitigating the effects of a pandemic and preparing for possible future pandemics [52,53]. However, the capacity issue of surgical wards was only one of the problems during the pandemic. Especially in the first months of the pandemic, the lack of rapid COVID-19 screening to prevent potential exposure to healthcare professionals and overwhelming numbers of COVID-19 patients disrupted hospital medicine and primary care workflows. Furthermore, the COVID-19 pandemic has shown that new oncological screening solutions are needed. In such a crisis, expanding online-based cancer screening tools [54] or utilizing AI-based tools [55] may prove helpful.

The authors acknowledge several limitations of this study. Firstly, GT does not provide a precise number of searches and only selects the data for the whole world or a whole country. It is also impossible to choose other macroregions (e.g., the continent). However, the RSV is adjusted to the number of Google users in the given region and time. Therefore, the index helps analyze the dynamics of discourse. Secondly, the topics’ RSV could often depend on media attention, which could be why some countries have a high amplitude of irregularity [56]. There is also a potential bias connected with the digital exclusion of the elderly or patients who lack routine Internet use.

## 5. Conclusions

The interest of Google users in cancer screenings increased in 2020–2021 compared to 2015–2019, but the growth was less dynamic than expected. The Relative Search Volumes of many cancers during the COVID-19 pandemic were significantly lower than in the prepandemic period, especially during March and April 2020. The lower interest in cancers and their screenings may delay the diagnosis of malignancies and worsen the long-term outcomes. When using data from GT, one should be aware of the most critical limitations of this tool: a limited selection of the analyzed regions and the significant impact of media attention on the level of interest in a given issue.

## Figures and Tables

**Figure 1 cancers-15-00617-f001:**
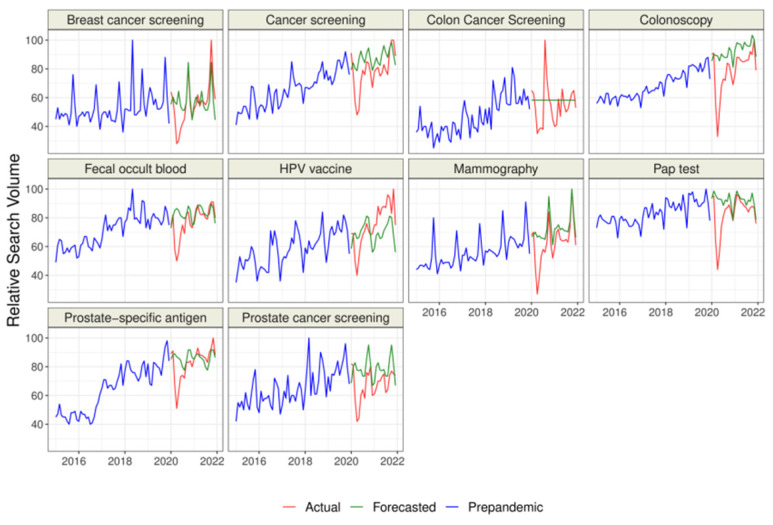
Relative Search Volume of Google queries on chosen cancer-related screenings.

**Figure 2 cancers-15-00617-f002:**
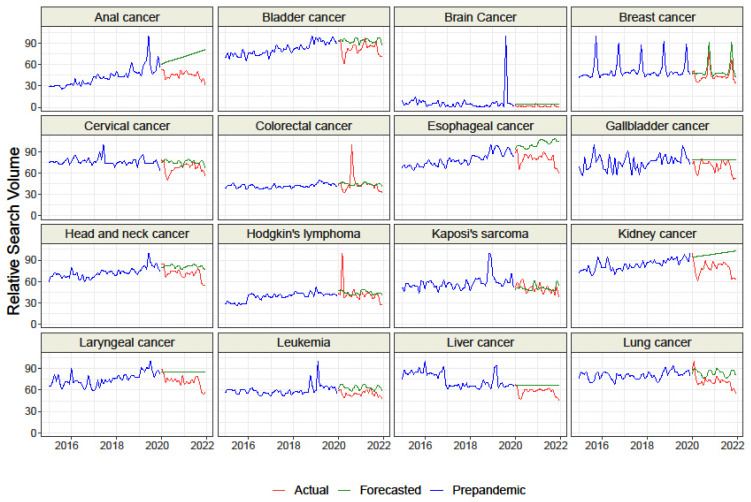
Relative Search Volume of Google queries on chosen malignancies.

**Figure 3 cancers-15-00617-f003:**
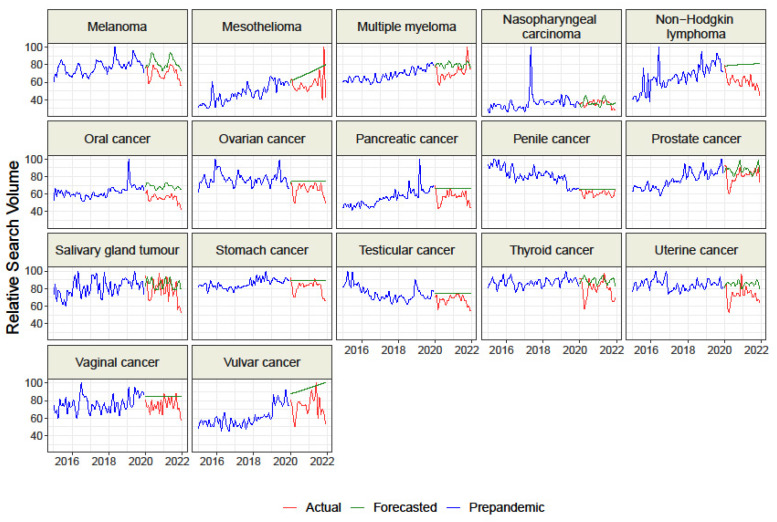
Relative Search Volume of Google queries on chosen malignancies.

**Table 1 cancers-15-00617-t001:** Global statistics of Google queries on the chosen cancer screenings and HPV vaccine.

Name of Screening-Related topic	Before Pandemic (RSV)	Forecasted Interest (RSV)	Actual Interest (RSV)	Difference: Actual vs. Forecast (RSV)	PrePandemic vs. during Pandemic * (*p*-Value)	Forecasted vs. Actual **(*p*-Value)
Breast cancer screening	50.0 (47.0–53.2)	56.2 (53.4–59.1)	56.0 (46.5–59.1)	−44.3	0.08	0.72
Cancer screening	67.0 (54.8–73.2)	87.1 (83.0–91.0)	78.5 (75.8–91.0)	−218.2	**<0.001**	**<0.01**
Colon Cancer Screening	43.0 (38.0–55.2)	58.3 (58.3–58.3)	54.0 (44.0–58.3)	−85.3	**0.025**	0.11
Colonoscopy	65.0 (61.0–74.5)	92.8 (88.6–96.6)	84.5 (72.8–96.6)	−334.0	**<0.001**	**<0.001**
Fecal occult blood	74.0 (61.0–79.2)	83.3 (80.8–86.5)	80.0 (74.5–86.5)	−130.2	**0.013**	0.07
HPV vaccine	58.5 (49.8–68.2)	68.6 (66.1–74.2)	75.0 (68.8–74.2)	129.2	**<0.001**	0.06
Mammography	54.0 (48.8–60.0)	71.0 (66.9–73.6)	63.5 (57.5–73.6)	−230.7	**<0.01**	**<0.01**
Pap test	82.0 (77.8–88.5)	92.8 (90.9–93.9)	87.0 (80.2–93.9)	−219.4	0.36	**<0.001**
Prostate-specific antigen	67.0 (47.0–77.2)	86.1 (84.2–88.5)	85.5 (78.5–88.5)	−82.9	**<0.001**	0.46
Prostate cancer screening	63.0 (56.0–74.0)	77.8 (73.5–82.1)	70.0 (60.8–82.1)	−264.1	0.20	**<0.001**

RSV—Relative Search Volume, * unpaired comparison of RSV before the COVID-19 pandemic vs. the actual RSV during the pandemic, ** unpaired comparison of RSV of the forecasted trend vs. the actual RSV during the pandemic.

**Table 2 cancers-15-00617-t002:** Global statistics of Google queries on the chosen malignancies.

Name of Malignancy	Before Pandemic (RSV)	Forecasted Interest (RSV)	Actual Interest (RSV)	Difference: Actual vs. Forecast (RSV)	PrePandemic vs. during Pandemic * (*p*-Value)	Forecasted vs. Actual **(*p*-Value)
Anal cancer	42.0 (32.0–48.2)	70.8 (66.2–75.7)	46.0 (41.0–75.7)	−625.1	0.09	**<0.001**
Bladder cancer	81.0 (76.0–88.0)	92.9 (91.8–95.6)	83.5 (76.8–95.6)	−266.7	0.57	**<0.001**
Brain Cancer	4.0 (2.0–7.0)	4.5 (4.5–4.5)	1.0 (1.0–4.5)	−70.6	**<0.001**	**<0.001**
Breast cancer	47.0 (45.0–50.2)	47.4 (46.5–49.0)	42.0 (40.0–49.0)	−167.7	**<0.001**	**<0.001**
Cervical cancer	75.0 (73.0–78.0)	75.0 (73.9–77.3)	68.0 (63.8–77.3)	−195.1	**<0.001**	**<0.001**
Colorectal cancer	41.0 (40.0–44.0)	44.6 (42.6–45.3)	43.0 (38.2–45.3)	−2.4	0.54	0.14
Esophageal cancer	77.0 (71.0–84.0)	100.2 (97.0–105.0)	81.0 (78.0–105.0)	−504.7	0.22	**<0.001**
Gallbladder cancer	75.0 (69.0–81.2)	78.8 (78.8–78.8)	71.5 (62.5–78.8)	−242.7	**0.010**	**<0.001**
Head and neck cancer	71.0 (67.8–75.2)	81.2 (79.3–82.5)	71.5 (67.5–82.5)	−245.3	0.82	**<0.001**
Hodgkin’s lymphoma	40.0 (37.0–42.0)	43.8 (41.7–46.3)	41.0 (37.0–46.3)	−51.3	0.44	**<0.01**
Kaposi’s sarcoma	57.0 (52.8–59.2)	49.9 (48.1–51.6)	49.5 (46.8–51.6)	−21.8	**<0.001**	0.54
Kidney cancer	84.0 (78.8–89.2)	98.3 (96.2–100.5)	79.5 (74.8–100.5)	−471.7	**0.036**	**<0.001**
Laryngeal cancer	74.5 (69.8–81.0)	85.2 (85.2–85.2)	72.0 (69.8–85.2)	−314.6	0.18	**<0.001**
Leukemia	58.0 (56.0–61.0)	63.1 (61.1–64.7)	54.0 (52.0–64.7)	−203.3	**<0.001**	**<0.001**
Liver cancer	69.5 (66.0–83.2)	66.8 (66.8–66.8)	60.0 (57.2–66.8)	−211.1	**<0.001**	**<0.001**
Lung cancer	81.0 (76.8–84.2)	84.1 (81.1–87.6)	72.0 (69.0–87.6)	−275.3	**<0.001**	**<0.001**
Melanoma	77.0 (70.0–81.0)	79.8 (77.7–84.1)	71.5 (64.8–84.1)	−282.7	**<0.001**	**<0.001**
Mesothelioma	47.0 (39.0–54.5)	70.1 (65.6–74.8)	55.0 (52.0–74.8)	−312.0	**<0.001**	**<0.001**
Multiple myeloma	67.5 (63.0–72.2)	79.9 (77.8–81.3)	69.5 (68.0–81.3)	−196.2	0.07	**<0.001**
Nasopharyngeal carcinoma	34.5 (31.0–38.0)	35.0 (34.3–36.9)	36.5 (34.5–36.9)	−18.4	0.30	0.57
Non-Hodgkin lymphoma	66.0 (57.8–73.2)	80.1 (79.4–80.7)	60.0 (55.8–80.7)	−464.1	0.05	**<0.001**
Oral cancer	61.0 (58.0–65.0)	68.5 (66.9–69.5)	56.0 (54.0–69.5)	−312.2	**<0.001**	**<0.001**
Ovarian cancer	76.5 (73.0–79.0)	75.0 (75.0–75.0)	66.0 (63.5–75.0)	−227.3	**<0.001**	**<0.001**
Pancreatic cancer	55.0 (48.8–61.0)	66.7 (66.7–66.7)	57.0 (55.2–66.7)	−244.2	0.30	**<0.001**
Penile cancer	81.0 (77.0–86.2)	65.9 (65.9–65.9)	61.0 (58.8–65.9)	−122.7	**<0.001**	**<0.001**
Prostate cancer	75.5 (68.8–83.0)	86.2 (84.8–89.6)	81.5 (76.0–89.6)	−161.2	**0.027**	**<0.01**
Salivary gland tumor	81.0 (74.0–88.0)	85.1 (79.5–87.0)	76.0 (70.0–87.0)	−167.7	0.32	**0.049**
Stomach cancer	84.0 (82.0–89.0)	89.2 (89.2–89.2)	83.5 (81.2–89.2)	−187.6	0.12	**<0.001**
Testicular cancer	72.5 (69.0–79.2)	75.1 (75.1–75.1)	67.5 (66.0–75.1)	−194.1	**<0.001**	**<0.001**
Thyroid cancer	87.0 (84.0–90.2)	89.6 (86.6–91.3)	81.0 (75.2–91.3)	−219.7	**<0.001**	**<0.001**
Uterine cancer	84.0 (80.0–87.0)	85.4 (83.2–86.3)	72.5 (69.2–86.3)	−289.6	**<0.001**	**<0.001**
Vaginal cancer	75.5 (70.8–82.2)	84.9 (84.9–84.9)	74.0 (70.8–84.9)	−244.8	0.48	**<0.001**
Vulvar cancer	58.0 (52.8–62.5)	93.8 (90.5–97.1)	74.5 (65.0–97.1)	−496.1	**<0.001**	**<0.001**

RSV—Relative Search Volume, * unpaired comparison of RSV before the COVID-19 pandemic vs. the actual RSV during the pandemic, ** unpaired comparison of RSV of the forecasted trend vs. the actual RSV during the pandemic.

## Data Availability

The data presented in this study are openly available at DOI.

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
