# Peer review of "Effects of the COVID-19 Pandemic on the Interest of Google Queries in Cancer Screening and Cancers: A Retrospective Study"

_cancers, 2023, doi:10.3390/cancers15030617_

Round 1
Reviewer 1 Report
The authors compared the interests of Google search users regarding various types of cancers and cancer preventions pre- and peri-pandemic periods. While the research ideas are interesting, I have a few comments to improve them further.
1. It would be helpful for readers if authors could provide regional variations in the RSVs to see if there are any specific countries or regions we need to work on to improve awareness of cancers.
2. The authors recommended improving surgical centers' capacity to mitigate the pandemic's effects on cancer prevention. However, what we saw during the pandemic on the frontline was not necessarily a capacity issue; they were more about a lack of rapid COVID-19 screening to prevent potential exposure to healthcare professionals, and overwhelming numbers of COVID-19 patients disrupted hospital medicine and primary care workflows. In this regard, I suggest authors consider other recommendations (such as expanding online-based cancer screening tools, utilizing AI-based tools, etc.).
Author Response
Dear Reviewer,
- Thank you very much for this valuable and very accurate comment. Such an analysis could be another exciting element for the reader. Unfortunately, this is quite a time-consuming and challenging analysis. We selected 42 topics for our analysis for only one region: worldwide. GT does not provide a quick method to generate time trends representing RSVs in different countries. Therefore, the number of time trends for analysis will increase proportionally to the number of countries chosen for regional analysis, e.g. for 5 countries; we need to download manually and analyze 5*42=210 time trends. That additional analysis will require more time than the main calculations presented in our paper. If, in your opinion, regional analysis is essential to accept this publication, we can do it, but unfortunately, we need more time than 10 days, e.g. 30 days.
- Thank you very much for your comment; we agree and feel that the relevant part should be added. We want to add it to the discussion along with the proper quotes: "However, the capacity issue of surgical wards was only one of the problems during the pandemic. Especially in the first months of the pandemic, a lack of rapid COVID-19 screening to prevent potential exposure to healthcare professionals and overwhelming numbers of COVID-19 patients disrupted hospital medicine and primary care workflows. Furthermore, the COVID-19 pandemic has shown that new oncological screening solutions are needed. In such a crisis, expanding online-based cancer screening tools [54] or utilizing AI-based tools [55] may prove helpful."
54. Koczwara, B.; Knowles, R.; Beatty, L.; Shepherd, H.L.; Shaw, J.M.; Dhillon, H.M.; Karnon, J.; Ullah, S.; Butow, P. Implementing a Web-Based System of Screening for Symptoms and Needs Using Patient-Reported Outcomes in People with Cancer. Support. Care Cancer 2023, 31, 69, doi:10.1007/s00520-022-07547-9.
55. Hunter, B.; Hindocha, S.; Lee, R.W. The Role of Artificial Intelligence in Early Cancer Diagnosis. Cancers 2022, 14, 1524, doi:10.3390/cancers14061524.
We have also made minor linguistic corrections in the text of the manuscript - these will be available in its new version. We have underlined the errors in red and crossed them out, and written the correct versions in red next to them.
Reviewer 2 Report
The article is interesting.
There are some limitations described, however in the conclusion section these are not reiterated. This aspect can be improved.
Author Response
Dear Reviewer,
Thank you very much for your comment; we agree and feel that the relevant part should be added. We want to add a sentence at the end of the conclusions summarizing the most important limitations: "When using data from GT, one should be aware of the most critical limitations of this tool: limited selection of the analyzed regions and the significant impact of media attention on the level of interest in a given issue."
We have also made minor linguistic corrections in the text of the manuscript - these will be available in its new version. We have underlined the errors in red and crossed them out, and written the correct versions in red next to them.
Reviewer 3 Report
minor language modifications are needed
Author Response
Dear Reviewer,
Thank you very much for your comment. We have made some linguistic corrections in the text of the manuscript - these will be available in its new version. We have underlined the errors in red and crossed them out, and written the correct versions in red next to them.
Round 2
Reviewer 1 Report
The authors appropriately addressed my concerns.